# Alloferon Affects the Chemosensitivity of Pancreatic Cancer by Regulating the Expression of SLC6A14

**DOI:** 10.3390/biomedicines10051113

**Published:** 2022-05-11

**Authors:** Hyejung Jo, Dahae Lee, Cheolhyeon Go, Yoojin Jang, Suhyun Bae, Tomoyo Agura, Jiye Hong, Dongmin Kang, Yejin Kim, Jae Seung Kang

**Affiliations:** 1Laboratory of Vitamin C and Antioxidant Immunology, Department of Anatomy and Cell Biology, Seoul National University College of Medicine, Seoul 03080, Korea; luv_jo@snu.ac.kr (H.J.); ddhh12345@snu.ac.kr (D.L.); rhcjfgus@snu.ac.kr (C.G.); pierce52@snu.ac.kr (Y.J.); baesh9706@snu.ac.kr (S.B.); tomoyoagura@gmail.com (T.A.); carpediem0912@naver.com (J.H.); 2Department of Psychological and Brain Sciences, College of Arts and Sciences, Boston University, Boston, MA 02215, USA; dong1109@bu.edu; 3Medical Research Center, Institute of Allergy and Clinical Immunology, Seoul National University, Seoul 03080, Korea

**Keywords:** alloferon, pancreatic cancer, SLC6A14, glutamine, gemcitabine

## Abstract

Pancreatic cancer (PCa), one of the most malignant solid tumors, has a high mortality rate. Although there have been many trials of chemotherapeutic drugs such as gemcitabine, the mortality rates remain significantly higher than for other types of cancer. Therefore, more effective ways of improving conventional therapy for PCa are needed. Cancer cells take up large amounts of glutamine to drive their rapid proliferation. Recent studies show that the amino acid transporter SLC6A14 is upregulated in some cancers alongside glutamine metabolism. Alloferon, a peptide isolated from the insect immune system, exerts anti-viral and anti-inflammatory effects via its immunomodulatory function. In addition, it has anti-tumoral effects, although the underlying mechanisms are largely unknown. Therefore, we investigated the effects of alloferon on the PCa cell lines Panc-1 and AsPC-1. Exposure of these cells to alloferon for 3 weeks led to the downregulation of SLC6A14 expression and decreased glutamine uptake. Given that SLC6A14 plays a role in tumor progression and survival by promoting glutamine uptake into cancer cells, alloferon could be a potential adjuvant for the chemotherapeutic drug gemcitabine.

## 1. Introduction

Pancreatic cancer (PCa) is an aggressive cancer with a high mortality rate and poor prognosis. Recently, many studies have been conducted on new treatments for PCa, but the mortality rate of PCa patients continues to increase [1]. According to statistics from the American Cancer Society, although it depends on the progression of the tumor, the 5-year overall survival of PCa in general is known to be 11%. However, if it has already progressed to distant metastasis, the 5-year overall survival is only about 3%. This suggests that the early detection and treatment of tumors are very important for the treatment of PCa [1,2,3]. Indeed, the incidence of PCa continues to increase across all regions and populations; however, only 10% of affected patients are eligible for surgery at the time of diagnosis because early diagnosis is challenging [3,4,5,6]. There have been numerous studies conducted to increase the survival rate of patients with cancers; these include cytotoxic chemotherapy, small molecule inhibitors, monoclonal antibodies, and programming of adaptive immunity; however, surgical excision is still the most successful method of treating PCa [4,7,8]. Gemcitabine is the most commonly used and effective chemotherapeutic agent for PCa. Unfortunately, recent developments in treatment and surgical techniques for PCa have not translated into improvements in the 5-year survival rate [6]. Therefore, improved therapeutic strategies based on gemcitabine are urgently needed.

Glutamine is an important amino acid that plays a crucial role in maintenance and promotion on cancer cell function. Glutamine metabolism is crucially important for cell proliferation and protein synthesis because the amino acid is used as a mitochondrial substrate for energy production and biosynthesis. It also provides nitrogen for nucleotide synthesis and activates mTORC1 in cells to regulate signaling pathways and maintain redox homeostasis [9,10,11,12,13,14,15,16,17]. The role of glutamine metabolism has been explored in breast cancer, liver cancer, lung cancer, ovarian cancer, and kidney cancer [18,19,20,21,22,23]. In fact, in some tumors, including PCa, a phenomenon called “glutamine addiction” shows that cancer cannot survive without exogenous glutamine supplementation [24,25]. These aforementioned studies suggest that potential therapeutic approaches should target glutamine metabolism in cancer cells.

Amino acid transporters are plasma membrane-bound proteins that selectively transport certain amino acids into and out of cells and their organelles [26,27]. Among these transporters, SLC6A14 interacts with a wide variety of substrates, including all neutral and cationic amino acids; the exceptions are glutamate and aspartate. The transporters use energy provided by Na^+^ and Cl^−^ gradients, as well as the membrane potential, to import amino acids into cells. SLC6A14 plays an important role in the uptake of glutamine and, therefore, in cancer cell survival [28,29]. SLC6A14 is upregulated in many different cancer types, including colon cancer, cervical cancer, estrogen receptor-positive breast cancer, PCa, and osteosarcoma [30,31,32,33,34,35,36].

Alloferon is a short peptide that was first extracted from the blood of *Calliphora vincina* larvae after bacterial challenge. There are two variants of alloferon, alloferon 1 and alloferon 2, and they differ in terms of the number of amino acids in the peptide chain: 13 and 12, respectively [37]. There are several reports regarding the role of alloferon as an anti-viral, anti-inflammatory, and anti-tumoral agent [38,39,40,41,42]. Previous studies show that alloferon has anticancer effects against prostate cancer and colorectal cancer; however, its effects against PCa are unknown.

Therefore, this study aimed to investigate the anticancer effects of alloferon as an adjuvant for gemcitabine. The results show that alloferon regulates the expression of SLC6A14 on PCa, thereby inhibiting glutamine uptake. Thus, alloferon is a promising new therapeutic strategy for PCa.

## 2. Materials and Methods

### 2.1. Cell Culture

The human PCa cell lines Panc-1 and AsPC-1 were purchased from the American Type Culture Collection (ATCC, Manassas, VA, USA). Panc-1 was cultured in DMEM (GE Healthcare, Chicago, IL, USA) supplemented with 10% heat-inactivated fetal bovine serum (FBS; HyClone, Logan, UT, USA) and antibiotics (100 U/mL of penicillin and 100 μg/mL of streptomycin; Welgene, Kyungsan, Korea). AsPC-1 was grown in RPMI1640 (GIBCO, Grand Island, NY, USA); this RPMI formulation has been modified by ATCC to include 10% FBS and 1% penicillin/streptomycin. For the experiments, these cell culture media were supplemented with 4 μg/mL of alloferon and added to cells for 3 weeks. Control cells were not treated with alloferon. The cell lines were incubated at 37 °C in a humidified incubator containing 5% CO_2_.

### 2.2. Immunofluorescence Microscopic Analysis

Panc-1 and AsPC-1 were treated for 3 weeks with or without alloferon (4 μg/mL) and then seeded onto 12 mm cover slips in a 24-well plate. Cells were then incubated overnight before being washed twice with phosphate buffered saline (PBS) and fixed for 15 min at 4 °C with 2% paraformaldehyde (PFA). For immunofluorescence staining, permeabilization solution (0.1% Triton X-100 in PBS) was added to the cells on coverslips and incubated for 15 min at room temperature (RT). To prevent nonspecific staining, washed cells were incubated for 60 min at RT with blocking buffer (0.1% Triton X-100, 0.5% BSA, 5% normal goat serum) and then exposed for 2 h at RT to rabbit anti-human SLC6A14 (MBL, Nagoya, Japan) diluted in blocking buffer. Then, the cells were incubated for 30 min at RT with Alexa Fluor 488-conjugated goat antirabbit IgG (Invitrogen, Carlsbad, CA, USA) diluted in blocking buffer. Cells were then mounted in mounting medium containing DAPI (Immuno Bioscience, Mukilteo, WA, USA). To do this, the coverslips were mounted face down on a slide glass onto which the mounting medium had been placed. Cells were analyzed under a fluorescence microscope (EVOS M5000; Invitrogen, Carlsbad, CA, USA). Mean fluorescence intensity was measured and normalized to the fluorescence signal generated by DAPI. 

### 2.3. Western Blot Analysis

Panc-1 and AsPC-1 were treated (or not) with alloferon (4 μg/mL) and then lysed in RIPA buffer containing 50 mM Tris-HCl (pH 7.4), 1% NP-40, 0.25% sodium deoxycholate, 150 mM NaCl, 1 mM EDTA, and a protease inhibitor cocktail. The protein concentration was measured in a BCA assay (Sigma, St. Louis, MO, USA). An equal amount of protein (30 μg/sample) was loaded into each lane of a 12% polyacrylamide-SDS gel and then separated by electrophoresis (100 V for 4 h). Proteins were then transferred onto a nitrocellulose membrane (400 mA for 1 h), which was then blocked for 1 h at RT with 5% nonfat milk in 0.1% Tween 20 in PBS (PBST). The blocked membrane was then incubated overnight at 4 °C with rabbit anti-human SLC6A14 (MBL) and mouse anti-human β-actin (Santa Cruz Biotechnology, Dallas, TX, USA) antibodies. After washing three times (each for 10 min) with PBST, the membrane was incubated for 1 h at RT with horseradish peroxidase (HRP)-conjugated antirabbit IgG (Cell Signaling Technology, Boston, MA, USA) to detect SLC6A14, or with HRP-conjugated anti-mouse IgG (Cell Signaling Technology) to detect β-actin. The membrane was then washed three times (each for 10 min), and the immunoreactive proteins were visualized with the Lumi Femto (DoGenBio, Seoul, Korea) detection system for SLC6A14 or the Lumi La (DoGenBio) detection system for β-actin. Band density was analyzed using Image J software (NIH, Bethesda, MD, USA). Results were expressed as intensity relative to β-actin.

### 2.4. CCK-8 Assay

Panc-1 and AsPC-1 cells were treated (or not) with alloferon (4 μg/mL) for 3 weeks and then seeded into a 96-well culture plate. After incubating overnight, cells were treated with 100 µL of gemcitabine (0.001, 0.01, 0.1, 1, 5, 10, 50, 100, 500, or 1000 µM) in complete medium with 10% FBS and then incubated for 72 h. To examine cell viability, the medium was aspirated, and 100 µL of cell proliferation and cytotoxicity assay kit (EZ-Cytox; Dogen) solution (10% EZ-Cytox solution in PBS) was added to each well. Absorbance was measured at 450 nm using a microplate reader and SoftMax Pro software (Molecular Devices). The IC50 of gemcitabine was determined using nonlinear regression analysis in GraphPad Prism 5 (GraphPad Software, La Jolla, CA, USA).

### 2.5. Glutamine Uptake Assay

Panc-1 and AsPC-1 cells were treated for 3 weeks with or without alloferon (4 μg/mL), seeded into a 6-well culture plate, and incubated overnight. Cells were then washed and incubated for 4 h in conventional culture medium without FBS. Then, the medium was changed to glutamine-free RPMI/DMEM to deplete glutamine. After incubation for another 4 h, the glutamine-free medium was replaced with RPMI1640 or DMEM containing 2 mM and 4 mM glutamine, respectively. The concentration of intracellular glutamine was measured in a glutamine assay kit (Promega, Madison, WI, USA). Briefly, cells were removed from plates using trypsin-EDTA and collected in a 15 mL conical tube. Cells were washed with PBS and lysed in 0.3 N HCI/450 mM Tris. The assay was performed according to the manufacturer’s instructions. Luminescence was measured using SoftMax Pro software (Molecular Devices, Sunnyvale, CA, USA).

### 2.6. Cell Cycle Analysis

Panc-1 and AsPC-1 were treated with or without alloferon (4 μg/mL) for 3 weeks and then seeded in a 100 cm^2^ culture dish. Cells were washed twice with PBS and then treated with 4 μg/mL of alloferon and gemcitabine. To determine that the maximum dose displayed a substantially lower alternation of cell cycle distribution, we treated gemcitabine within a concentration range from 0 to IC50 in media (Panc-1: 0, 0.25, 0.5, 1, 10 μM; AsPC-1: 0, 0.0625, 0.125, 0.25, 5 μM). After incubation for 72 h, cells were harvested using trypsin-EDTA. Next, cells (5 × 10^5^) were fixed in 75% ethanol (added drop by drop) and incubated overnight at −20 °C. Fixed Panc-1 and AsPC-1 cells were washed with PBS and FACS buffer (PBS containing 0.5% BSA and 0.1% sodium azide). For staining, cells were resuspended for 15 min at RT in 0.5 mL of propidium iodide (PI)/RNase staining buffer (BD Pharmingen, San Diego, CA, USA) and analyzed by flow cytometry using a FACSLyric^TM^ cytometer (BD Biosciences, San Jose, CA, USA). Data were analyzed using FlowJo software (BD Biosciences, Franklin Lakes, NJ, USA).

### 2.7. Statistical Analysis

Data are presented as the mean ± SD. An unpaired two-tailed t test was used to compare two groups. *p*-values < 0.05 were considered statistically significant. Statistical analysis was carried out using GraphPad Prism 5 (GraphPad Software).

## 3. Results

### 3.1. Alloferon Suppresses the Expression of SLC6A14 in PCa Cell Lines

Based on two of previous reports showing the upregulation of SLC6A14 in PCa cells [34,35] and our previous report regarding the anti-tumor activity of alloferon against tumor cells through the activation of NK cells [41], we examined whether alloferon affects its direct anti-tumor activity through SLC6A14 expression on Panc-1 and AsPC-1. After cells were treated with 4 μg/mL of alloferon, changes in SLC6A14 expression were assessed at weeks 1, 2, and 3 by immunofluorescence microscopy. As shown in Figure 1A, SLC6A14 expression on Panc-1 fell after 2 weeks, with the decrease being even more pronounced at 3 weeks. Similarly, SLC6A14 expression on AsPC-1 fell markedly at 2 weeks (Figure 1B). The immunofluorescence data were confirmed by Western blot analysis. SLC6A14 expression decreased after alloferon treatment (Figure 2). In addition, we found that the basal expression of SLC6A14 was higher in Panc-1 than in AsPC-1 (Appendix A). Taken together, the data suggest that alloferon inhibits the expression of SLC6A14 in PCa cells. Also, based on these data, we treated cells with 4 μg/mL alloferon for 3 weeks prior to all future experiments.

### 3.2. Alloferon Increases the Chemosensitivity of Panc-1 and AsPC-1

Next, we investigated changes in the chemosensitivity of Panc-1 and AsPC-1 cells by measuring the IC50 against gemcitabine in the presence/absence of alloferon. A previous study suggests that blockade of SLC6A14 triggers alternative metabolic pathways involving tryptophan, branched-chain amino acids, ketone bodies, and membrane phospholipids, and that dysfunction of SLC6A14 plays an important role in increased chemosensitivity [43]. After culture for 3 weeks in the presence or absence of alloferon (4 μg/mL), cells were exposed to gemcitabine (from 0.001 nM to 1000 μM) for 72 h. The average IC50 value in Panc-1 decreased in the presence of alloferon (without alloferon: 11.83 ± 1.47 μM; with alloferon: 9.22 ± 1.01 μM). The same was true for AsPC-1 (without alloferon: 4.04 ± 1.54 μM; with alloferon: 3.12 ± 0.39 μM) (Figure 3). Thus, alloferon-treated PCa cell lines tended to be more chemosensitive than untreated control cells. However, these results suggest that alloferon does increase the chemosensitivity of PCa cells to gemcitabine.

### 3.3. Alloferon Inhibits Glutamine Uptake in PCa Cells

As we observed that the expression of SLC6A14 falls after treatment with alloferon, we performed a glutamine uptake assay to examine whether glutamine uptake of Panc-1 and AsPC-1 also decreased. The assay scheme is shown in Figure 4A. First, we found that the absolute intracellular glutamine concentration in Panc-1 cells (5.54 ± 0.48 μM, Figure 4B) was higher than that in AsPC-1 cells (1.12 ± 0.08 μM, Figure 4C) in the absence of alloferon. After culture of Panc-1 and AsPC-1 cells for 3 weeks in the presence of 4 μg/mL alloferon, we observed a decrease in the intracellular glutamine concentration in Panc-1 (3.72 ± 0.34 μM after treatment, Figure 4B) cells. However, there was no difference in intracellular glutamine concentration in AsPC-1 cells in the presence of alloferon (1.05 ± 0.14 after treatment, Figure 4C). Therefore, we next examined changes in the glutamine uptake ability of cells after exposure to alloferon after 4 h of incubation in glutamine-free culture medium, followed by another 4 h of incubation in culture medium containing glutamine.

When cells were incubated for 4 h in glutamine-free culture medium, the intracellular glutamine concentration in Panc-1 cells in the presence of alloferon fell to 0.84 ± 0.48 μM (compared with 1.68 ± 0.07 μM in the absence of alloferon) (Figure 4B). Interestingly, although the intracellular glutamine concentration in Panc-1 in the absence of alloferon recovered rapidly (4.75 ± 0.26 μM) when the cells were exposed to glutamine-containing medium, recovery was much lower in cells cultured with 4 μg/mL of alloferon for 3 weeks (Figure 4B). By contrast, there was no significant difference in intracellular glutamine concentration in AsPC-1 cells in the presence/absence of alloferon; indeed, we found extremely low intracellular glutamine concentrations in AsPC-1 even in the absence of alloferon (Figure 4C). Therefore, the data suggest that alloferon alters glutamine uptake in some PCa cells.

### 3.4. Co-Treatment of PCa Cells with Alloferon and Gemcitabine Alters Cell Cycle Distribution

Because the sensitivity of Panc-1 and AsPC-1 cells to gemcitabine increased after exposure to alloferon, we performed cell cycle analysis to investigate the effects of alloferon on the cell cycle distribution in the presence of gemcitabine. First, experiments were conducted to determine the highest concentration of gemcitabine that did not show cytotoxic effects. No cytotoxic effects of gemcitabine against Panc-1 and AsPC-1 cells were observed at 0.0625 μM and 0.5 μM, respectively (Appendix A). Combined treatment with gemcitabine and alloferon increased the percentage of apoptotic Panc-1 cells in the sub-G1 phase (2.1 ± 0.3% in the absence of alloferon vs. 3.5 ± 0.3% in the presence of alloferon, *p* = 0.02; Figure 5A). However, alloferon had no synergistic effect on the sub-G1 population of AsPC-1 cells (Figure 5B). Thus, alloferon increases the percentage of Panc-1 cells in the sub-G1 phase.

## 4. Discussion

The cause of PCa is still largely unknown, and there are still no effective treatments (other than early resection) that extend survival [2]. Currently, many studies aim to maximize the effects of gemcitabine as a PCa treatment by combining it with other anti-cancer drugs. However, increased drug toxicity and the induction of cancer cell resistance are the biggest barriers [44,45,46,47]. Therefore, this study aimed to examine the adjuvant effects of alloferon, a peptide drug originating from the insect immune system and which is known to have potential anticancer effects [40,41].

First, we investigated whether alloferon alters the expression of SLC6A14, which is highly expressed in PCa cells and is closely related to metastasis, proliferation, and resistance to chemotherapeutic drugs [34]. As expected, the expression of SLC6A14 on PCa cell lines Panc-1 and AsPC-1 was downregulated markedly by treatment with alloferon for 3 weeks, even though the expression did not change within the first week (Figure 1 and Figure 2). Interestingly, we found that the expression of SLC6A14 was higher in Panc-1 than in AsPC-1, and that changes in glutamine uptake and cell cycle status in response to alloferon were more significant for Panc-1 than for AsPC-1. Based on the role of SLC6A14 in cancer cell metabolism and growth, and the finding that Panc-1 is more malignant than AsPC-1 [48,49], the data suggest that alloferon could be used as a chemo-adjuvant drug to regulate metastasis and the proliferation of more aggressive cancers. Glutamine is essential for proliferation, protein synthesis, and chemoresistance in cancer cells; thus, cancer cells are said to suffer “glutamine addiction” [9,17,24,25]. Glutamine, taken up by SLC6A14, is used as a mitochondrial substrate for energy production and biosynthesis. Moreover, glutamine metabolism fuels the tricarboxylic acid cycle, nucleotide and fatty acid biosynthesis, and redox balance in cancer cells. In addition, glutamine activates mTOR signaling, thereby reducing stress on the endoplasmic reticulum and promoting protein synthesis [50,51,52,53]. Therefore, the suppression of glutamine uptake via the downregulation of SLC6A14 expression may increase the efficacy of cancer chemotherapy [32,33,35,54,55,56].

As shown in Figure 4, alloferon decreased glutamine uptake in Panc-1 cells, as well as reducing the intracellular concentration. This suggests that alloferon might be an effective suppressor of “glutamine addiction”. The main reason for comparing the change of SLC6A14 expression with the change in glutamine uptake by the treatment of alloferon in both Panc-1 and AsPC-1 is that Panc-1 is highly resistant to gemcitabine, while AsPC-1 is more susceptible to gemcitabine. In other words, the degree of expression of SLC6A14 in PCa against gemcitabine was thought to be a factor for chemosensitivity to gemcitabine. The high expression of SLC6A14 in Panc-1 and the resulting increase in glutamine uptake are thought to be the cause of high resistance to gemcitabine. As shown in Figure 2A,B, it can be confirmed that the relative expression of SLC6A14 is significantly lower in AsPC-1 than in Panc-1. It implies a direct correlation with a lower amount of glutamine uptake in AsPC-1 compared to Panc-1 (Figure 4). Since the expression amount of SLC6A14 is basically significantly lower in AsPC-1 than in Panc-1, although SLC6A14 was reduced in AsPC-1 by alloferon (Figure 1B, Figure 2B, and Appendix A), it is considered that there would be no significant difference in the amount of glutamine taken.

Next, we examined whether the chemosensitivity of PCa cell lines to gemcitabine increases after exposure to alloferon for 3 weeks. In the absence of alloferon, the IC50 of gemcitabine in Panc-1 was higher than that in AsPC-1 (11.83 ± 1.47 μM vs. 4.04 ± 1.54 μM, respectively; Figure 3). Moreover, the IC50 decreased after treatment with alloferon (Panc-1: 9.22 ± 1.01 μM; AsPC-1: 3.12 ± 0.39 μM). The difference in the IC50 for the two cell lines is again thought to be due to differences in SLC6A14 expression. Nevertheless, the results suggest that alloferon may increase the chemosensitivity of PCa to gemcitabine, particularly PCa cells expressing high levels of SLC6A14.

Since gemcitabine exerts cytotoxic effects by blocking DNA synthesis, we examined changes in cell cycle distribution after treatment with gemcitabine in the presence or absence of alloferon. PCa cell lines were exposed to noncytotoxic concentrations of gemcitabine after pre-treatment with alloferon for 3 weeks. The population of Panc-1 cells in sub-G1 increased slightly but significantly (2.1 ± 0.3% before vs. 3.5 ± 0.3% after, *p* = 0.02; Figure 5). Even though we have found a minor increase in dead cells in the sub-G1 stage from 2.1–3.5 percent and no significant change in AsPC-1 cells, we additionally performed a TUNEL assay to confirm this change. As shown in Appendix A, TUNEL positive cells have been found in both “Gemcitabine” and “Alloferon + Gemcitabine”, but there was no statistical significance. It is consistent with the results for glutamine uptake and the IC50 of gemcitabine; we found no meaningful changes in the cell cycle distribution of AsPC-1 cells. Again, this is likely due to the fact that the expression of SLC6A14 in AsPC-1 cells is markedly lower than that in Panc-1 cells. Therefore, it should be further clarified through experiments about the in vivo effect of alloferon and gemcitabine using the xenograft model.

Taken together, the results presented herein suggest that alloferon might be a useful adjuvant for gemcitabine-based chemotherapy for PCa, particularly PCa that expresses high levels of SLC6A14 and is highly resistant to gemcitabine. However, further investigations using other gemcitabine-resistant PCa cells are needed to confirm these data. The limitation of the study is that the combination effect on alloferon and gemcitabine was confirmed only through the cell cycle.

Although SLC6A14 is a powerful glutamine transporter, other amino acid transporters such as SLC1A5, SLC38A1, SLC38A3, SLC48A5, and SLC7A5 play important roles [28,29]. Indeed, downregulation of SLC6A14 inhibits the function of SLC7A11 and SLC7A5 [29]. Thus, the effects of alloferon on the expression of other amino acid transporters should be investigated to clarify whether it blocks the uptake of other essential amino acids required by cancer cells.

## 5. Conclusions

In summary, this study demonstrated that (1) alloferon effectively decreases the expression of the glutamine transporter SLC6A14 in pancreatic cancer cells, (2) thereby reducing the uptake of glutamine into the cells, and (3) decreased intracellular glutamine concentration might increase sensitivity to gemcitabine and increase apoptotic cells. This is the first study regarding the treatment of pancreatic cancer using alloferon as an adjuvant of conventional chemotherapeutic drugs. Therefore, the present study provides evidence for a new potential use of alloferon and gemcitabine for the treatment of pancreatic cancer by the regulation of SLC6A14 expression.

## Figures and Tables

**Figure 1 biomedicines-10-01113-f001:**
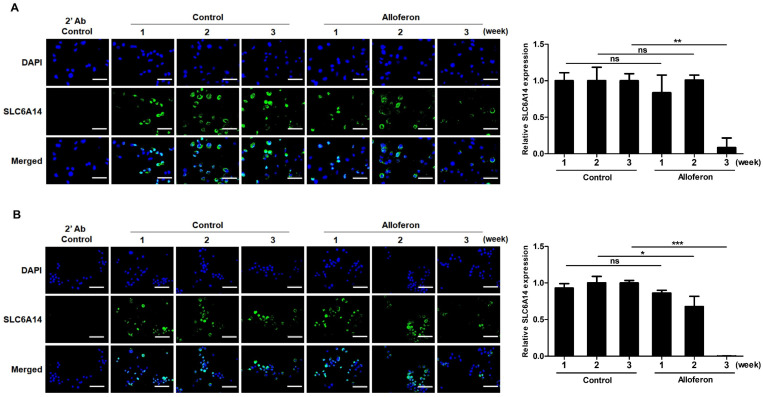
Immunofluorescence microscopic analysis of changes in expression of SLC6A14 by pancreatic cancer cell lines Panc-1 and AsPC-1 upon exposure to alloferon. Panc-1 and AsPC-1 were cultured for 3 weeks in the presence or absence of alloferon (4 μg/mL). Changes of SLC6A14 expression in control (untreated) and alloferon-treated cells were examined after 1, 2, and 3 weeks. SLC6A14 protein was quantified by measuring GFP fluorescence (green), and nuclei were stained with DAPI (blue): (**A**) Panc-1 and (**B**) AsPC-1. 2’Ab control means that it is stained only with secondary antibody. Images are shown at 400× magnification (scale bar, 125 μm). Expression of SLC6A14 at different times was normalized to that of SLC6A14 expression in control Panc-1 and AsPC-1 during the first week (set as 1). Data are presented as the mean ± SD. * *p* < 0.05; ** *p* < 0.01; *** *p* < 0.001; ns = not significant.

**Figure 2 biomedicines-10-01113-f002:**
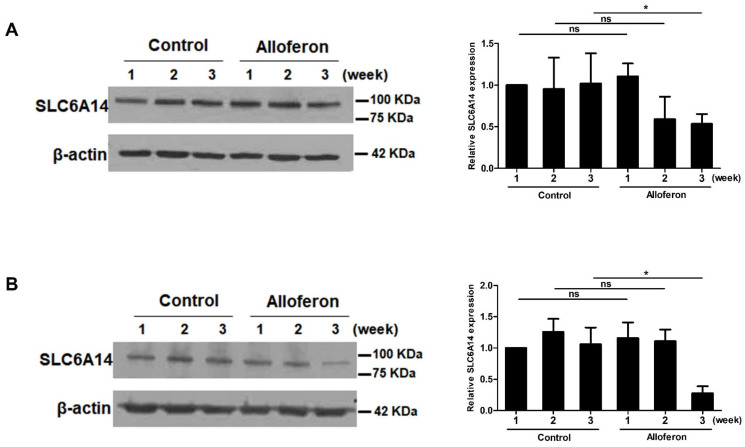
Western blot analysis of changes in SLC6A14 expression by pancreatic cancer cell lines Panc-1 and AsPC-1. Cells were cultured for 3 weeks in the presence or absence of alloferon. Western blot analysis was performed to examine SLC6A14. β-actin was used as a loading control: (**A**) Panc-1, (**B**) AsPC-1. Relative changes in SLC6A14 expression are shown in the bar graph. Results are representative of five independent experiments. Data are presented as the mean ± SD. * *p* < 0.05; ns = not significant.

**Figure 3 biomedicines-10-01113-f003:**
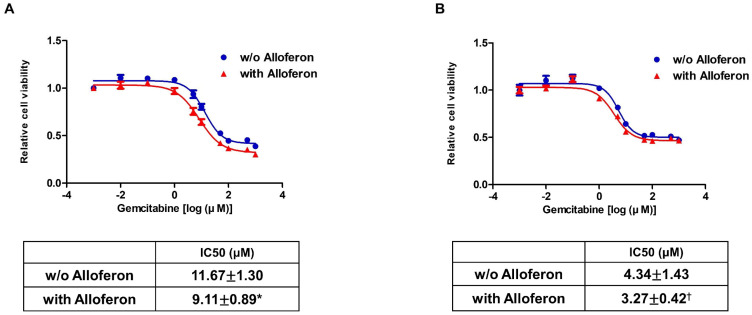
Alloferon reduces the IC50 of gemcitabine against pancreatic cancer cells. Cells were cultured for 3 weeks in the presence or absence of 4 μg/mL of alloferon and then seeded in a 96-well culture plate. Cells were treated with gemcitabine (0.001 nM to 1000 μM) and then incubated for 72 h. Cell proliferation was normalized to that of cells treated with 0.001 nM gemcitabine: (**A**) Panc-1 and (**B**) AsPC-1. Results are representative of four independent experiments. Data are presented as the mean ± SD. * *p* < 0.05 for Panc-1 (**A**); ^†^
*p* = 0.12 for AsPC-1 (**B**).

**Figure 4 biomedicines-10-01113-f004:**
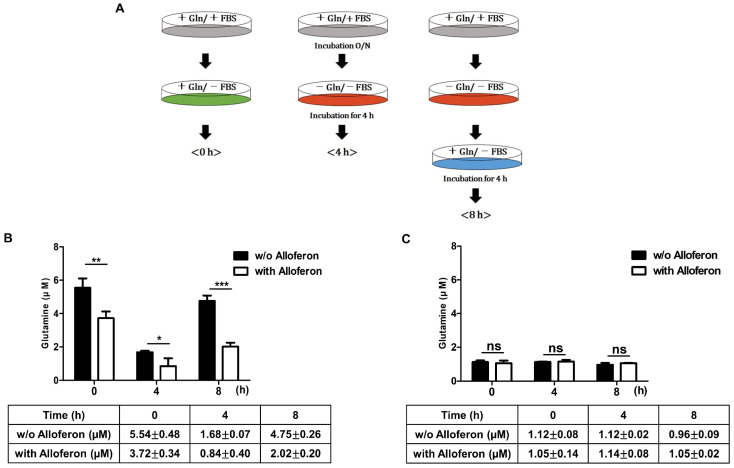
Changes in intracellular glutamine levels in pancreatic cancer cell lines exposed to alloferon. Panc-1 and AsPC-1 cells were grown overnight in complete medium containing 2 or 4 mM of glutamine, respectively. Cells were then cultured for another 4 h in glutamine-free medium. Intracellular glutamine concentration before and after the addition of exogenous glutamine was then measured. (**A**) Schematic showing the glutamine uptake assay. The intracellular glutamine concentration was measured using a glutamine assay kit. (**B**) Panc-1 cells and (**C**) AsPC-1 cells. Each sample was tested in triplicate, and data are presented as the mean ± SD. * *p* < 0.05; ** *p* < 0.01; *** *p* < 0.001; ns = not significant.

**Figure 5 biomedicines-10-01113-f005:**
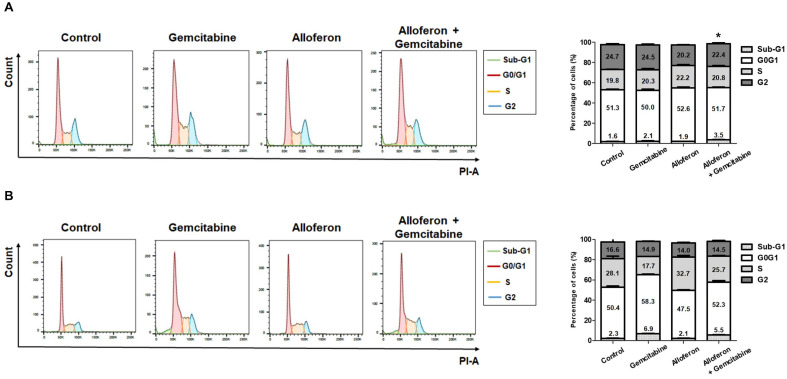
Changes in the cell cycle distribution of Panc-1 and AsPC-1 cells exposed to alloferon or gemcitabine. After treatment with or without alloferon for 3 weeks, Panc-1 and AsPC-1 cells were seeded into a 100 cm^2^ culture dish. Cells were then treated with 4 μg/mL alloferon, gemcitabine (Panc-1: 0.5 μM; AsPC-1: 0.0625 μM), or a combination of both for 72 h, followed by PI/RNase staining and flow cytometry analysis: (**A**) Panc-1 and (**B**) AsPC-1. The percentage of cells in each phase of the cell cycle is shown in the bar graph. Results are representative of three independent experiments, and the data are presented as the mean ± SD. * *p* < 0.05 (sub-G1 Gemcitabine vs. sub-G1 Alloferon + Gemcitabine).

## Data Availability

All datasets generated for this study are included in the article or Appendix A.

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
