# Peer review of "Alloferon Affects the Chemosensitivity of Pancreatic Cancer by Regulating the Expression of SLC6A14"

_biomedicines, 2022, doi:10.3390/biomedicines10051113_

Round 1

Reviewer 1 Report

I have raised few issues related to the study, I hope authors could work on the comments to make it interesting and meaningful. My comments are mentioned below:

1. First line of the abstract and introduction is exactly similar. Please reframe any of them. Authors need to mention in the abstract about overall survival rate or 5-year survival rate. For more information see: https://doi.org/10.3892/ijo.2021.5290

  1. In the introduction section, authors need to mention about the statistics by American cancer society related to pancreatic cancer. Please do cite the recent epidemiological statistics about pancreatic cancer such as in the introduction section of article https://doi.org/10.1177/1533033820962117.
  2. Authors in their introduction section states "Previous studies show that alloferon has anticancer effects against prostate cancer and colorectal cancer". Could authors provide the related reference ? Additionaly, it will be interesting to know how alloferon exhibit its anticancer activity and mechanism in these cancer types.
  3. In figure 2, authors in the figure legend stated "Relative changes in SLC6A14 expression are shown in the bar graph". I assumed it to be related to alloferon treatment and not the vehicle control. Please do analyze and add the graph for control group as well in order to have a clear picture. It should look like what is shown in the bar graphs of figure 1.
  4. In figure 2A, western blot for SLC6A14 from Alloferon treatment does not seems to be a representative and matching from the bar graph. SLC6A14 expression looks over-expressed in 2 week compared to 1 week (Alloferon treated). Please replace the more accurate representative blot corresponding to the bar graph.
  5. In figure 3, authors please do mention about the condition of the medium while undergoing gemcitabine treatement. i.e. complete medium with 10%FBS or 1%FBS or serum-free medium? Since IC50 value difference ranges from 1-2μM (with or without alloferon) it is really important to know about the treatment medium used.
  6. Authors in the section 3.4 stated “However, alloferon had no synergistic effect on the bub-G1 population of AsPC-1 cells (Figure 5B)”. Please do correct bub-G1 to sub-G1. Surprisingly, in combined treatment, apoptotic cell death is only increased from 2.1% to 3.5% and based on these small changes it is not justified to state that alloferon synergize or augment the effect of Gemcitabine. It is recommended to perform ISOBOLOGRAM analysis. For more information on isobologram analysis please read: https://doi:10.3389/fphar.2019.01222, and https://doi.org/10.1158/1535-7163.MCT-10-0606
  7. In figure 5, there is no B panel and there is no information about C panel. Authors please proof read the entire figure 5 (including panels A to C) and its associated text in the result section.
  8. Authors found minor increase in the apoptotic cell death or sub-G1 peak and p-value is not mentioned anywhere. Please do mention p-value for combined treatment in text or figure. With no p-value I doubt about a change. Hence, it is highly recommended to perform microscopy based tunnel assay or western blot based PARP cleavage or caspase-3 activation (subject to availability for any one of the experiment).
  9. In the discussion section, Authors stated “However, we found no significant changes in uptake of glutamine or its intracellular concentration in AsPC-1 cells (Figure 4C). This may be due to the much lower basal expression of SLC6A14 in AsPC-1 (Supplemental 297 Figure S1)”. This entire statement doesn’t hold true as figure 1B shows detectable and quantifiable expression of SLC6A14 in Aspc1 cells. Authors need to state more relevant argument to discuss.
  10. It is strange to see that alloferon is decreasing the levels of SLC6A14 in AsPC-1 cells (figure 1B) but it is not able to effect glutamine dependency of these cells. This strange result somehow decreases the importance of the entire manuscript. Authors need to be careful while discussing this. Authors please do find a probable answer to this question by searching the literature in depth. This issue needs to be included in the discussion section.

Reviewer 2 Report

Paper well written. Just a minor comment. The authors should implement the discussion reporting the limitations of the study. Furthermore, the statement reported should be mitigated considering the initial phase of the study of the Alloferon effects  

Round 2

Reviewer 1 Report

In response to my previous comments, authors has have improved the manuscript quality. Still the manuscript is not suitable for publication in the present lacking form. I hope authors could work on the new comments to make it more scientifically suitable for publication.

Comments are as follows:

1.) Response to my comment 1. I must appreciate authors for incorporating the valuable information on pancreatic cancer in introduction section. (Thanks, Issue is resolved).

 2.) Response to my comment 2. I must appreciate authors for incorporating the valuable statistics on pancreatic cancer in introduction section. (Thanks, Issue is resolved).

3.) Response to my comment 3. In response to my previous comment number 3 "Could authors provide the related reference related to anticancer effects of alloferon against prostate cancer and colorectal cancer", authors provided reference number 38 and 39 which are Kuczer et al 2010 and Kuczer M et al 2013 respectively. Unfortunately, both these references does not correspond according to the authors statement in response letter. Reference 38 shows effects of Alloferon on VERO cells and reference 39 shows anti-inflammatory effects of alloferon on HaCaT cells.

Therefore I again request authors to quote the correct references and give the previously published studies the due credits. (Issue Un-resolved).

4.) Response to my comment 4. I appreciate the explanation of authors related to calculation and normalization of band for SLC6A14. Everyone knows how to calculate such values. Authors provided the new bar graph, issue resolved. However, my this particular comment was inspired from the present day scrutinizing of western blots data (published data) on peer review websites. Factually the representative blots shown in articles should be corresponding to the bar graphs. (Thanks, Issue is resolved).

5.) Response to my comment 5. I must appreciate authors for incorporating my suggestion. (Thanks, Issue is resolved).

6.) Response to my comment 6. I must appreciate authors for incorporating my suggestion. (Thanks, Issue is resolved).

7.) Response to my comment 7. Authors stated "So, as shown in Figure 3, we first examined whether Alloferon can reduce IC50 of Gemcitabine. Of course, …………….. the main purpose of our study is to decrease the toxicity and increase the efficacy of gemcitabine by treatment of alloferon. For this reason, IC50 of gemcitabine was checked first, and then the effect of alloferon as as an adjuvant for gemcitabine was examined". I have some serious concern related to the figure. Without stating the p-value and such a small change this entire figure does not hold true. Authors please provide p-values for the IC50 values stated in the table. (Issue Un-resolved).

8.) Response to my comment 8. I must appreciate authors for incorporating my suggestion. (Thanks, Issue is resolved).

9.) Response to my comment 9. This is really an important issue as alloferon increases the percentage of Panc-1 cells in sub-G1 phase from 2 – 3.5 percent and no significant change in AsPc1 cells. The entire conclusion of the manuscript seems objectionable. It is very unfortunate to know that only 10 days were provided, whereas alloferon treatment stands out for 3 weeks in itself. But it is not justifiable to show non-significant changes as a basis for the entire manuscript. So, I suggest authors to ask for some extra time and perform TUNEL assay or WB for PARP cleavage or Caspase-3 activation as stated previously. With so much of delusional data, in vivo settings might not work. Please do the needfull (Issue Un-resolved).

10.) Response to my comment 10. I must appreciate authors for incorporating my suggestion. (Thanks, Issue is resolved).

11.) Response to my comment 11. I must appreciate authors for incorporating my suggestion. (Thanks, Issue is resolved).

Round 3

Reviewer 1 Report

1.) In response to my previous comment 3, regarding references of line 166-168 "Previous studies show that alloferon has anticancer effects against prostate cancer and colorectal cancer; however, its effects against PCa are unknown". Although authors provided the appropriate reference but they forgot to correct or edit the lines in revised manuscript. Authors are again requested to provide reference number 41 at the end of line 168. Additionally edit the line according to the response stated by authors. (Issue still un-resolved).

2.) In response to my comment 7, regarding mentioning of p-value for figure-3. Thanks for providing the p-values. But from the fig legend it is not clear which p-value correspond to which cell line.

Therefore I suggest to mention " *P < 0.05 for Panc-1 (A); †P = 0.12 or non-significant for Aspc (B)". (Issue partially solved).

3.) In response to my comment 9, regarding tunnel assay. Thanks for performing and providing the Tunnel assay. Even though the results remain non-significant and no change, I highly suggest to provide this data as a supplementary figure with minor description in result and/or discussion section. In my opinion this data could serve as a basis for pointing out limitation of the present study and a reason for your future in vivo studies. I can only only recommend the publication of the manuscript if this experiment is properly provided and discussed in the manuscript. (Issue partially resolved).

I hope and highly expect authors future work on alloferon to be successfull in mice studies.
